# Estimating Information Processing of Human Fast Continuous Tapping from Trajectories

**DOI:** 10.3390/e24060788

**Published:** 2022-06-04

**Authors:** Hiroki Murakami, Norimasa Yamada

**Affiliations:** 1Graduate School of Health and Sport Sciences, Chukyo University, 101 Tokodachi, Kaizu-cho, Toyota, Aichi 470-0393, Japan; 2School of Health and Sport Sciences, Chukyo University, 101 Tokodachi, Kaizu-cho, Toyota, Aichi 470-0393, Japan; nyamada@sass.chukyo-u.ac.jp

**Keywords:** Fitts, information entropy, mutual information

## Abstract

Fitts studied the problem of information capacity and transfer in the speed–accuracy motor paradigm using a theoretical approach developed from Shannon and Weaver’s information theory. The information processing (bit/s) estimated in Fitts’ study is calculated from the movement time required to achieve the required task index of difficulty but is essentially different from Shannon’s information entropy. Thus, we estimated the information entropy of multiple human movement trajectories and the mutual information among trajectories for the continuous aiming task in Fitts’ paradigm. Further, we estimated the information processing moment by moment. Two methods were considered: (1) encoded values encompassing the coordinates of the three dimensions and (2) coordinate values associated with each direction in the three dimensions. Information entropy indicates the magnitude of variation at each time point, and the structure of this variation varies with the index of difficulty. The ratio of entropy to mutual information was examined, and it was found that information was processed from the first half of the trajectory in difficult tasks. In addition, since these values calculated from the encoded method were higher than those from the conventional method, this method may be able to estimate these values successfully.

## 1. Introduction

Fitts [1] was the first to empirically study the problem of information capacity and transfer in the velocity–accuracy motion paradigm. His study was a theoretical approach developed from the information theory of Shannon and Weaver [2]. He described the human motor system from the equation MT=a+b ID by setting an index of difficulty (ID; $ID=\log_{2}\frac{2A}{W}$) determined from the distance to the target (A) and the target width (W). This ID has a unit of information (bit), and the above equation indicates that when the ID is higher, the movement time (MT) required to process more bits is larger. This equation also known as Fitts’ law has been recognized as the most important basic principle of human physical movement [3].

Information entropy is a functional measure corresponding to information capacity in human information processing models [4] and is used to infer uncertainty, disorder, and variability in physical systems [2,5,6]. It has also been applied to human motion. For example, the Kolmogorov–Sinai entropy [7] and approximate entropy [8] methods have been applied to time series with strong periodicities, such as those found in basic human motion patterns. This was used to quantify the regularity of the data. However, in the approach of Fitts [1], developed from information theory, information entropy is not calculated, and information capacity is estimated from the ID determined by the range of variation of endpoints [1,9]. In light of this, an attempt has been made to link the concept of information entropy which is common in terms of uncertainty, with the concept of the ID in Fitts’ law [10]. In other words, exercise outcome was measured directly in terms of probability. For example, using the characteristics of the frequency distribution, the probability is calculated from the amount of data corresponding to each bin, the information entropy is calculated, the distribution of the raw data is assumed to be Gaussian, and the information entropy is calculated by finding the probability at each time point of the trajectory [10]. However, in all these studies, discrete aiming movements were used. Furthermore, the analysis was conducted using data only in the main direction of motion, with data acquired by moving in a plane in two dimensions. Notably, this is not a problem since Fitts’ law has been adapted to discrete motion in his later studies [11]. However, one possible reason this method is still used is that it is difficult to analyze whether the motion is continuous in three dimensions. Additionally, since Fitts’ experiment [1] used a continuous aiming motion which is a three-dimensional motion, it was considered necessary to verify the results following the original Fitts experiment.

In a different vein, our previous study [12], using trajectories toward the target in Fitts’ experimental paradigm as the topic of analysis, showed that the values (variation values) quantified by principal component analysis for the three-dimensional coordinates corresponding to any time point in the trajectory gradually decreased toward the target. We thought that since Fitts’ original study [1] did not fully control for the possibility of deviation from the target that occurred during the trial and also reported that a 4% error was allowed to occur [13], it was necessary to study this in an environment where the error was controlled to 0%. This study adds a new condition wherein the error rate is controlled to 0% and shows that in such cases, this decrease is even greater than in the previous condition. In addition, this new study examined how the variability of the trajectory toward the target changes when the error is controlled to 0%. Furthermore, the study concluded that the gradual decrease in the magnitude of variability toward the target may have been the result of information processing. In other words, rather than the result of information processing appearing at the target endpoint, this study expressed the process of information processing before reaching the target. However, although that analysis used three-dimensional coordinates of the raw data, it did not follow changes in all data because we used a principal component analysis to transform the data into one dimension and then examined changes in those values, as in other previous studies with one-dimensional data as the subject of analysis. We have not been able to examine information processing from the perspective of probability because of information entropy, as in the aforementioned study by Lai et al. [10].

Therefore, this study examined information processing from a probability perspective by estimating directly from the concept of information entropy the possibility that information processing was performed to reduce variation based on the reduction in variation described above. In doing so, we attempted to estimate entropy using the coordinates obtained from the three-dimensional motion as they are. In addition, given that it has been pointed out that the trajectories of hand movements such as those in this study differ from one individual to another [14], we considered it necessary to first examine the characteristics of each participant separately. Therefore, as a first step before examining the generality of information processing, we decided to examine the results of two IDs (easy and difficult conditions defined by Fitts’ ID) from a large number of trials by a single participant.

## 2. Materials and Methods

### 2.1. Participants

One right-handed healthy man enrolled in the researchers’ university participated in the study. The participant self-reported that he had normal or corrected-to-normal vision and no motor impairments. All study procedures were conducted in accordance with the Declaration of Helsinki and the ethics code of Chukyo University and were approved by the ethics committee of Chukyo University (approval number: 2018–2029). The participant provided written informed consent before participation.

### 2.2. Apparatus

A motion-capture system (MAC3D System, Motion Analysis Corporation, Rohnert Park, CA, USA, 200 Hz) with six 300,000-pixel cameras arranged around the participant was controlled by software (Cortex version 7, Motion Analysis Corporation, Rohnert Park, CA, USA) to acquire the coordinates of the reflection markers attached to the stylus tip. The three-dimensional axes were set in the software such that the *x*-axis was in the direction of the distance between the targets (Figure 1). The accuracy of the analysis was 0.03 ± 0.28 mm of residual from the reference value in a space of 1.8 × 1.8 × 1.8 (m^3^) in the calibration performed prior to the experiment.

### 2.3. Experimental Design

The participant was instructed to tap two targets alternately, as accurately and quickly as possible, for 15 s. In addition, following our previous study [12], the error rate of the tap was controlled to 0% by setting the target as a 15 mm table. The participant did not receive any instructions regarding their body posture. The ID of the tasks was determined by fixing the width of the target at 15 mm and varying the target center-to-center distances (A: 60 and 480 mm). The equation used to determine the ID was: $ID=\log_{2}\frac{2A}{W}$; in this experiment, there could be two IDs obtained from this equation (ID: 3.00, 6.00). In addition, the participant was asked to self-report whether they needed more rest between trials to ensure that that they had enough time to rest. The participant was able to self-report and had sufficient rest between the trials.

### 2.4. Experimental Procedure

The participant entered the laboratory, sat down on the chair in front of the desk with the target, and was briefed on the task and risks of the experiment. The participant was verbally instructed to place the stylus tip on the right side of the target and wait for the signal to start the experiment and tap accurately and as fast as possible. First, the participant conducted 10 trials as a practice task for each ID and then conducted 100 trials for each ID. The start and end signals were the start sound of the motion-capture software at the beginning of the capture and the end sounds at the end of the capture (15 s after the start), respectively. No feedback was provided to the participant regarding the right number of taps.

### 2.5. Data Analysis

The error rates for each trial were calculated using the same definition as that of Fitts [1,11]. This was done to check whether the error rate was controlled at 0%, as in a previous study [12]. The errors in each trial were visually checked by the experimenter and confirmed using the captured data.

Following previous studies [15,16], the time corresponding to the peak and valley values was alternately calculated using the time-series coordinate data of the *x*-axis of the stylus tip. Furthermore, the time difference was determined as the MT for each trial. In other words, the MT was obtained for the number of taps.

Using the time-series coordinates in the *x*-axis direction regarding the direction of movement, the velocity and acceleration were calculated by first- and second-order differentiation, respectively, and both were smoothed with a cutoff frequency of 3 Hz. The local extreme values in Fitts’ experiment [12,17,18], i.e., peak velocity, peak acceleration, and peak deceleration (ID6), were calculated as the percentage of each MT. The characteristics of local extreme values of acceleration or deceleration differed for each ID. The position was negative when the target was tapped to the left and positive when the target was tapped to the right. For ID 3, the time-series acceleration data were similar to a sine wave, and one local extreme value, which was the peak acceleration, was calculated for each tap (Figure 2B, black square). Conversely, the time-series data of acceleration in ID6 have three localities. First, the instant at which the target is tapped can be calculated by examining the data together with the mutation data (Figure 2A, black circles). Next, there is a local extreme value immediately after the displacement begins from this tap, which is the peak acceleration (Figure 2A, black square). Shortly before the next tap, there is a local extreme value, which is defined as the peak deceleration (Figure 2A, grey square).

Using the time of all taps, we determined 11 equal time points (i.e., 0% MT, 10% MT, …, 100% MT) from each MT and the three-dimensional coordinates of these points in turn [19]. In other words, 11 time points were obtained sequentially for each tap, indicating that multiple coordinates were obtained at each time point for each trial (Figure 3).

The basis of information entropy is probability [2]. In previous studies, the probability of data distribution at each time point of the trajectory was used to calculate the information entropy for discrete aiming motions [9,10,20]. We believe that the same analysis as in the previous study could be performed for a continuous aiming motion, such as in the present study, by analyzing the data distribution at each time point for trajectories moving in the same direction. For example, the coordinates of each time point in the trajectory obtained in this study for the right-to-left target are shown in Figure 4.

In this study, the probabilities required to compute information entropy were obtained using two methods. In the first method, entropy was calculated by encoding the space of a regular cube with 15 mm as one side for the coordinates of each time point based on a target size of 15 mm per side. In other words, by giving multiple 15 mm regular cubes in the three-dimensional space where the experiment was conducted, we calculated which cube the three-dimensional coordinates corresponded to at each time point and determined the probability of this at each time point. To simplify the analysis, the coordinates in the *z*-axis direction of the target point were standardized by subtracting the average value of the target endpoint from the measured value for each trial. For example, in this analysis, the value of the information entropy is zero if the coordinates of all taps in 100 trials fall within the same cubic space (encoded in the same value). In our previous study [12], the error rate was controlled to 0% when a height-added table was used for the target. Therefore, the entropy at the target endpoint was theoretically expected to be zero. However, practically, due to errors in motion-capture measurements, it was expected to be slightly larger than zero. The second method calculates entropy according to the frequency distribution of the coordinates at each time point as in previous studies [9,10]. This is calculated using $\sum P_{i}\log_{2}\left(1/P_{i}\right)$, where $P_{i}$ is the frequency distribution of the data points in each bin *i*. This analysis was computed at each time point by coordinates in each of the three axes (*x*-, *y*-, and *z*-axis) of the three dimensions. This was calculated by fixing the bin width to 15 mm.

The coordinates of each point in time were used to analyze the amount of mutual information. Mutual information is defined as a measure of the amount of information that one random variable contains regarding another random variable [2,5]. Specifically, it quantifies the average amount of common information between two different time points and reveals the sequential interrelationship of two different variables to reveal the relative uncertainty of later values when the encoded values and trajectory locations at an earlier time point are known [5,10]. For example, consider the case of finding mutual information at 90% MT and 100% MT points of the trajectory. Since the probabilities at the two points in time are $P_{90}$ and $P_{100}$, and the joint probability is $P_{90,100}$, the mutual information is calculated from the following equation: $I_{90,100}=P_{90}+P_{100}-P_{90,100}$ [21]. In general, large mutual information indicates that the two variables have a high level of common information, meaning that uncertainty is greatly reduced. Contrarily, if mutual information is zero, it means that the two variables are independent of each other [21]. In this study, in addition to information entropy, this mutual information was also calculated using two methods: The first method uses the encoded values for the three-dimensional coordinates used to calculate entropy, and the second method uses the coordinates for each direction in the third dimension.

### 2.6. Statistical Analysis

The MT was calculated at each tap in each trial, and an average was calculated for each trial. The mean and standard deviation for all trials were calculated for each ID. Error rates were calculated for each trial and totaled for each ID. Peak velocity, peak acceleration, and peak deceleration as percentages of time were calculated for each tap in each trial, and the average was calculated for each trial. The mean and standard deviation for all trials were calculated for each ID. The MT, peak velocity, and peak acceleration as percentages of time were compared between IDs using the *t*-test.

Entropy and mutual information were calculated for trajectories moving in the same direction (trajectories moving from the left target to the right target and from the right target to the left target), and the entropy at each of the 11 time points for each ID and mutual information between each time point was averaged. This analysis averaged out the differences between the left and right human movements.

## 3. Results

### 3.1. Error Rate and Movement Time

Using a 1.5 cm high table as the target for tapping, similar to the results of a previous study [12], the error rate for all trials was 0%. This indicates that, although there was some variation within the target, there was no deviation from the target, and the participant was able to perform trials under both conditions while maintaining the accuracy of the tap. The movement time (MT) was 0.223 ± 0.030 for ID 3 and 0.621 ± 0.033 for ID 6. A significant difference was observed between the two IDs (*t* (198) = 98.568, *p* < 0.001).

### 3.2. Analysis of Extreme Value and Phase Diagram

Three representative extreme values (peak velocity, peak acceleration, and peak deceleration) have been recognized in previous studies for each trajectory when moving from target to target [17,18]. Figure 5 shows the position–velocity and position–acceleration phase diagrams for each ID. This phase diagram uses all trajectories during the trial for a typical example of each ID, allowing us to observe qualitative changes in velocity and acceleration. The ratio of the peak velocity to MT was 53.187 ± 1.242 for ID3 and 44.008 ± 1.068 for ID6, indicating a significant difference between the two IDs [*t* (198) = 56.034, *p* < 0.001]. The percentage of peak acceleration relative to MT was 3.096 ± 1.273 for ID3 and 19.271 ± 1.315 for ID6, with a significant difference between the two IDs [*t* (198) = 88.378, *p* < 0.001]. Finally, the percentage of peak deceleration relative to MT observed only in ID6 was 66.214 ± 2.021.

### 3.3. Information Entropy and Mutual Information

The results of the information entropy calculated by encoding the three-dimensional coordinates at each of the 11 time points are shown in Figure 6A, and the normalized values are shown in Figure 6B. Because all taps are at the target point, the value of entropy at the target point (0%MT, 100%MT) is theoretically zero, but it was not zero (Figure 6A). This is because, as expected, they were perceived to have deviated from their targets in the analysis. For example, the total number of taps on the left target in 100 trials was 3279 for ID3, of which 113 (3.1%) were recognized as deviations from the target in the analysis. For ID6, the value increased toward the midpoint of the trajectory, reached a maximum value (50%MT: 3.20) at the midpoint and then decreased again toward the target (100%MT). Conversely, for ID3, the values were lower at all time points than those for ID6. The normalized data have the largest value as 1 and the smallest value as 0, so the relative relationship between the values at each time point can be examined. Comparing ID6 and ID3, the time point indicating the maximum value was different (ID6:50%MT, ID3:20%MT), but the time point indicating the minimum value was the target point (ID6:0% and 100%MT, ID3:0% and 100%MT), with similar trends.

Next, using the coordinates for each axis direction, the results of calculating the entropy at each of the 11 time points, with the bin width fixed to the length of one side of the target size (15 mm), are shown in Figure 7A,C,E. The normalized values are shown, along with the encoding method (Figure 7B,D,F). The *x*-axis value related to the direction of movement in ID6 increased toward the midpoint of the trajectory and decreased again toward the target, as in the value for the encoding method (Figure 7A). In ID3, as with the encoding method, the values were lower at all time points than those in ID6 and showed little variation in the range of 0 to 1. The values in the *y*- and *z*-axis directions changed little at any point in time for both ID3 and ID6 (Figure 7C,E). However, it can be seen that there are many points in time where the value is slightly larger in ID6 than in ID3. This indicates that the variation is slightly larger in ID6, even for the axes that are not directly related to the direction of movement. The normalized data have the largest value as 1 and the smallest value as 0, so the relative relationship between the values at each time point can be examined. The characteristic result was that the values for all axes of ID6 showed a maximum at approximately the midpoint (40–60% MT) and a minimum at approximately the target point (0%, 100% MT).

The mutual information was calculated using two methods, as in entropy analysis. A relative comparison of the results for ID3 and ID6 reveals some general trends. First, the results of calculating the mutual information between different time points in the trajectory using the encoded values of the three-dimensional coordinates of each time point are shown in Figure 8. The structure of the change in mutual information at adjacent time points is similar to that of the entropy change. For ID6, the mutual information at the midpoint of the trajectory gradually increased from the beginning until it reached a maximum and then gradually decreased toward the end (Figure 8A). In addition, the values of mutual information at adjacent time points were the highest after 50% vs. for ID3; the values were generally lower and less variable even in the mutual information, similar to the structure of entropy change (Figure 8B).

Subsequently, we show the results of calculating the mutual information using the method in which the bin width is fixed to the target size (15 mm). Note that Figure 9 shows only the values calculated from the *x*-axis related to the direction of movement. In this analysis, the overall amount of mutual information for ID6 is greater than that for ID3. As a result, the structure of the change in the mutual information at adjacent time points was similar to the structure of the change in the values calculated by the encoded method and thus is considered the same as the structure of entropy.

## 4. Discussion

The purpose of this study was to directly quantify the variability of trajectories during a continuous-aiming task using Fitts’ experimental paradigm in terms of information entropy and mutual information content to estimate the amount of trajectory information processing. In doing so, the three-dimensional coordinates of each of the 11 time points (0%, 10%, …, 100% MT) of the trajectory were directly transformed into a measure of information entropy, a concept of probability, by introducing and encoding the same scale as the target point in three-dimensional space and fixing the bin width in each axis. The method used in this study to encode the three-dimensional coordinates of each time point allows for the consideration of influences other than the axis of the direction of movement, which is primarily important in Fitts’ experimental paradigm [1]. Prior studies have addressed movement in two or three dimensions as the subject matter, and only one-dimensional data, the direction of movement, are taken into account in the analysis [9,10,20]. Most studies conducted using Fitts’ experimental paradigm dealt with coordinates related to the direction of movement in data analysis of trajectories [10,17]. Our previous study [12] obtained coordinates in three dimensions, which is not an exception because this value was treated as one-dimensional data by principal component analysis. In this study, to facilitate the calculation of probabilities for each time point, we fixed the width, length, and height of the target to 15 mm and the size of the reflective marker and manipulated two IDs (ID = 3, 6) by changing the distance between the targets. We then encoded the three-dimensional coordinates of each time point into the cube by giving the experimental space a regular cube of 15 mm on one side. This is a newly devised method for calculating probabilities from variables that encompass the influence of three-dimensional coordinates. In contrast to this method, the probability of each bin at each time point was calculated using a method that fixed the width of the bin at 15 mm on each axis.

Information entropy obtained using experimental data may provide a more accurate estimate of information entropy than the assumed structure of the data distribution or the entropy of an a priori defined task [9,10]. In this study, the entropy values obtained from the coded methods and *x*-axis coordinates in ID6 increased from the beginning of the trajectory (0% MT), reached a maximum at the midpoint (40% MT and 50% MT), and then decreased toward the end (100% MT). The structure of this entropy change corresponds to the velocity profile in Figure 5A, where the average velocity peak point (44% MT) and entropy peak point (40% MT–50% MT) are nearly identical. This supports the results of previous studies, which indicated that movement velocity is a mediator of trajectory entropy [10]. The decrease in the value of information entropy in the latter half of the trajectory may be related to feedback control of limb movements [22]. Previous studies have analyzed the time up to the velocity peak as the feedforward in the initial impulse phase and the time from the velocity peak to the target as feedback in the current control phase [17,22]. Therefore, from the velocity peak, which is the point where the information entropy reaches its maximum, feedback control is initiated toward the target endpoint, which may have decreased entropy. Furthermore, the onset of peak deceleration in ID6 occurred on average at 66% of the trajectory, and the entropy value also decreased significantly from the 70% point toward the target (100% MT) (Figure 6A,B). This trend was similar to the results of our previous study [12], wherein the variability of trajectories toward the target (60–100% MT in this study) was quantified by principal component analysis. In other words, entropy may decrease significantly, particularly from the point where the acceleration reaches its extreme in the feedback control. This means that in addition to the results supporting the claim that changes in the value of information entropy are difficult to separate from the kinematic analysis of temporal changes in trajectory [10], the analysis of changes in information entropy must consider the extreme value points of velocity and acceleration. Conversely, the entropy values obtained from the encoded methods and *x*-axis coordinates in ID3 did not change significantly from the beginning of the trajectory (0% MT) to the end (100% MT), with only minor fluctuations. Moreover, the structure of entropy changes in ID3 was not consistent with the velocity profile. At the peak velocity point (53% MT), the entropy value is lower than at any other point in time except for the start and endpoints. In contrast to ID6, the time-series waveform of the acceleration at ID3 was sinusoidal. This is consistent with the findings of previous studies observed under low-ID conditions [23,24,25] and reveals that one extreme point (peak acceleration) is obtained per movement, which, on average, occurs in the early stages of the trajectory.

The values of entropy in the *y*- and *z*-axis directions did not change from values close to 0 from the beginning (0% MT) to the end (100% MT) of the trajectory for both ID6 and ID3, and the fluctuations were slight. The structure of change in the encoded entropy and the entropy in the *x*-axis direction were almost identical, indicating that the effect of the direction of movement is highly relevant to the effect of the variation at each time point. However, the value of entropy obtained by encoding three-dimensional coordinates is higher at each time point than the value of entropy obtained from coordinates along the *x*-axis. Therefore, rather than calculating the value of entropy separately for each axis, it is possible to provide more accurate values at each time point by considering a measure that includes information for all axes, suggesting that this may be a possibility. This is because it may ignore changes in other axes that have slight variations.

Mutual information analysis was performed to reveal the sequential interrelationships between the variables at two different time points [10]. This estimate of mutual information complements the value of information entropy at different points in the trajectory [10]. In this study, this mutual information was calculated using two methods, including information entropy. Further, the overall value was higher for ID6 than for ID3, which may reflect the predetermined ID of the task. This trend was observed in both analyses using both methods. However, as in the analysis of information entropy, the value of mutual information calculated from the value of the encoded method is higher than that of the other method. This suggests that a similar examination of a measure containing information on all axes may provide a more accurate amount of mutual information. A comparison of adjacent time points in Figure 8 and Figure 9 also reveals that the structure is similar to the pattern of change in information entropy in ID6. It has been suggested that just as the extremes of velocity and acceleration are related to changes in the value of information entropy, they may also be related to changes in the amount of mutual information [10]. However, the fact that the mutual information of the two adjacent time points in ID6 is higher at the midpoint of the trajectory (30–60% time points) and lower at the first and second half time points (0–30% and 60–100% time points) is probably because it depends on the magnitude of the probability value at a single time point. Specifically, since the information entropy at 0% and 100% time points is small, it is natural that the mutual information, including these points, is small. Conversely, in ID3, where the entropy values at each time point are low, the mutual information is low overall. We consider that this is not a problem in terms of analysis when discussing differences in the amount of information between IDs, but it is detrimental when discussing from which point the information processing took place within the same ID. When we compare two time points, mutual information may have the same value regardless of whether the uncertainty is decreasing (from a large value of entropy to a small value) or not (from a small value of entropy to a large value). Therefore, we consider relative information processing of ID6 by determining the ratio of the amount of information shared by two adjacent time points (mutual information) to the amount of entropy at a later time point. For example, if one wants to determine the percentage of mutual information between the 90% and 100% time points in the entropy at 100%, the equation is $\frac{I_{90,100}}{P_{100}}$.

The results of this analysis are shown in Figure 10 for the values of the encoded methods analyzed, with ID6 having a ratio of more than 0.5 the 10–20% MT point, indicating a higher percentage of mutual information in the entropy at later points in time. From this analysis, we considered the possibility of estimating information processing independent of the magnitude of entropy at each time point. In other words, information processing to reduce uncertainty occurred in ID6 from the time of peak acceleration until the target was reached [12]. This result further extends the possibility of information processing for trajectories, focusing only on the second half of the trajectory of our previous results, suggesting that information processing takes place from the first half of the trajectory.

Furthermore, similar to the results of previous studies [9,10], the peak value of the information entropy obtained at each time point in this study was less than the ID theoretically estimated by Fitts [1] based on the target width and distance. This suggests a limitation of Fitts’ approach, determined by the assumed structure of the data distribution and the a priori defined average information given the claims of previous studies, and suggests that estimates of information entropy based on experimentally obtained data may have been more accurately measured [9,10]. However, we must be careful when comparing our method with the information estimated by Fitts because we modified the following two points from Fitts’ original experiment to define the endpoint size for the calculation of entropy in a straightforward manner. First, we controlled the error rate to 0% from the method of our previous study. Fitts’ experiment assumed that the distribution of target endpoints follows a Gaussian distribution; therefore, he specified the width of the target by allowing for an error of approximately 4% [1,26]. In this study, no errors were made from the target. However, if experiments were to be conducted in an error-prone environment, the value of information entropy at the target point would be larger. Secondly, the target has no length in the *y*-axis direction; Fitts [1] used a *y*-axis length of 6 inches. This means that Fitts’ experiment allowed more target point redundancy than the present study. When experiments are conducted in such an environment, the value of the information entropy at the target point is expected to be larger, and a new rule for the size to be encoded, which was used in the analysis of this study, should be considered. In the future, it will be necessary to examine the generality of various participants and changing conditions.

## 5. Conclusions

In this study, we used Fitts’ experimental paradigm to quantify trajectory variability during a continuous aiming task as values of information entropy and mutual information content to estimate the amount of trajectory information processing. Entropy indicates the magnitude of variability at each time point, and we found that the structure of this variability varied with the ID of the task. The ratio of entropy to mutual information at later time points showed that information processing occurred from the first half of the trajectory in the difficult task. In addition, the estimation of information entropy and mutual information values may be provided more accurately using a measure that encompasses information in all dimensions of task movement.

## Figures and Tables

**Figure 1 entropy-24-00788-f001:**
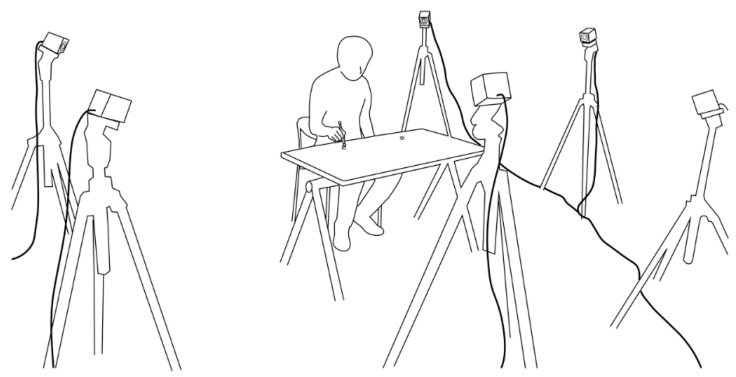
Conceptual diagram of experiment setup.

**Figure 2 entropy-24-00788-f002:**
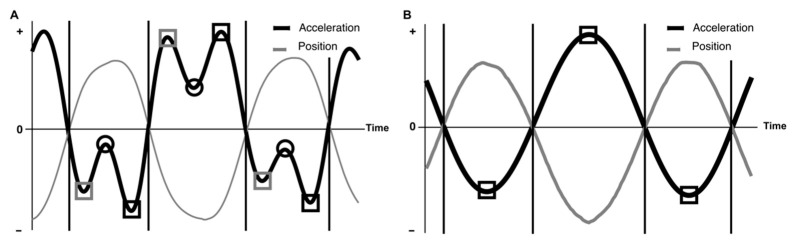
Conceptual diagram of local extrema value search method. (**A**) Typical time series data of normalized acceleration and position for ID6. (**B**) Typical time series data of normalized acceleration and position for ID3. The plot shows the normalized acceleration and position series observed in four half-cycles of movement divided into three segments (each segment ranged from one zero-crossing in position (movement midpoint) to the next zero-crossing), each containing a movement reversal.

**Figure 3 entropy-24-00788-f003:**
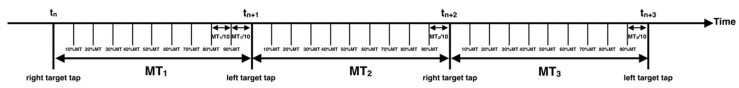
Conceptual diagram of how each time point is determined. First, let {t_n_, t_n+1_, t_n+2_, t_n+3_, …} be the times of the taps. Thus, the time between taps is MT, which is calculated sequentially. The three-dimensional coordinates of that time are {{t_n_, x_n_, y_n_, z_n_}, {t_n+1_, x_n+1_, y_n+1_, z_n+1_}, {t_n+2_, x_n+2_, y_n+2_, z_n+2_}, {t_n+3_, x_n+3_, y_n+3_, z_n+3_}, …}. Subsequently, by dividing each MT into 10 equal parts, each time point (10%MT, …, and 90% MT) is calculated, and the 3D coordinates of the time are calculated in order. The coordinates of each time point toward the target in the same direction are summarized.

**Figure 4 entropy-24-00788-f004:**
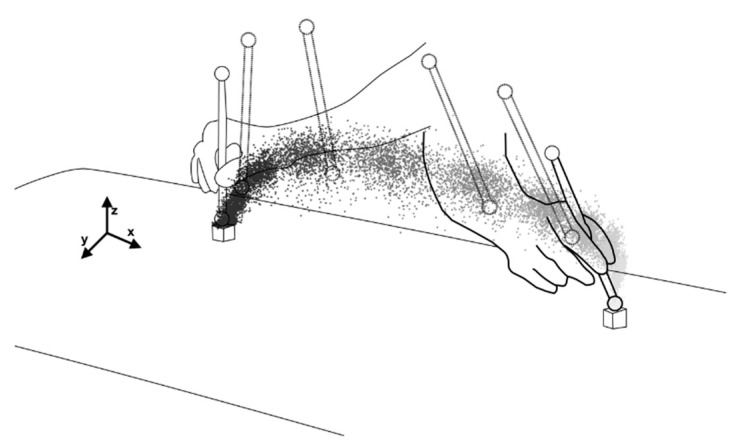
Conceptual diagram of the trajectory and coordinate system. The position of the stylus tip indicates the approximate position at each time point (0%, 20%, 40%, 60%, 80%, 100%).

**Figure 5 entropy-24-00788-f005:**
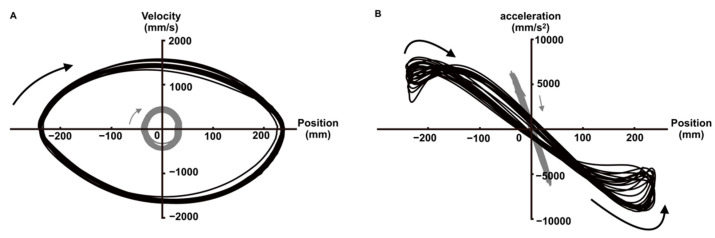
(**A**) Phase diagram of a typical position–velocity for each ID condition (black: ID6, gray: ID3). (**B**) Phase diagram of a position–acceleration for each ID condition (black: ID6, gray: ID3). The arrows show the qualitative characteristics of velocity and acceleration when moving from the left target to the right target.

**Figure 6 entropy-24-00788-f006:**
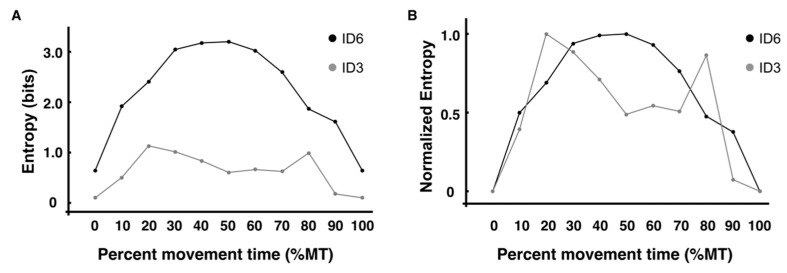
Information entropy is calculated by encoding the three-dimensional coordinates. (**A**) The entropy values of each time point are indicated by each ID. (**B**) The normalized entropy values of each time point are indicated by each ID.

**Figure 7 entropy-24-00788-f007:**
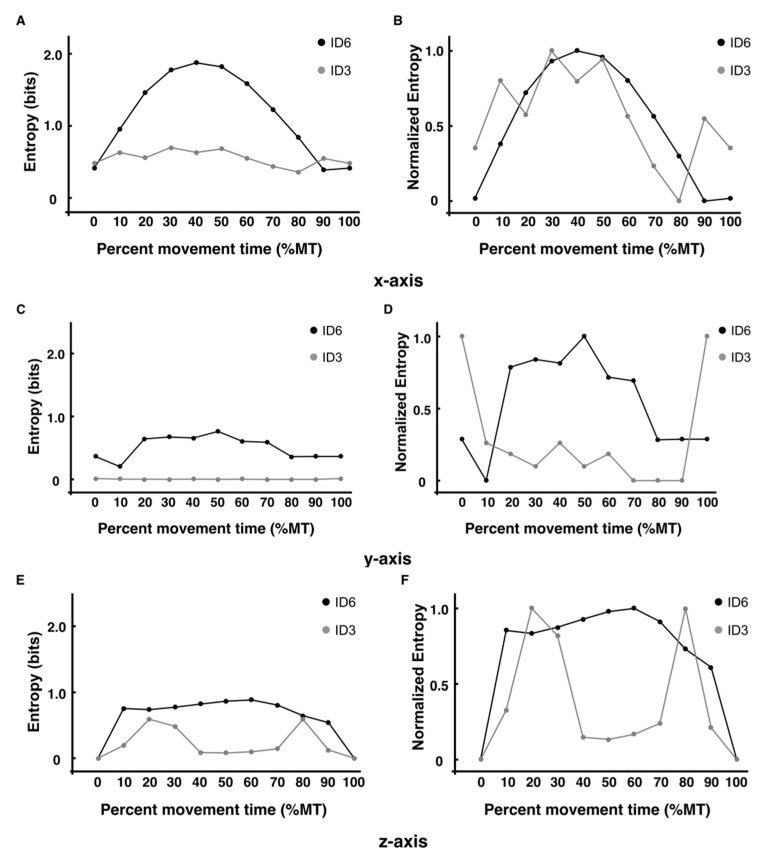
The entropy of each axis direction (*x*, *y*, *z* coordinates). (**A**,**B**) The *x*-axis entropy values and normalized entropy values of each of the 11 time points are indicated by each ID. (**C**,**D**) The *y*-axis entropy values and normalized entropy values of each time point are indicated by each ID. (**E**,**F**) The *z*-axis entropy values and normalized entropy values of each time point are indicated by each ID.

**Figure 8 entropy-24-00788-f008:**
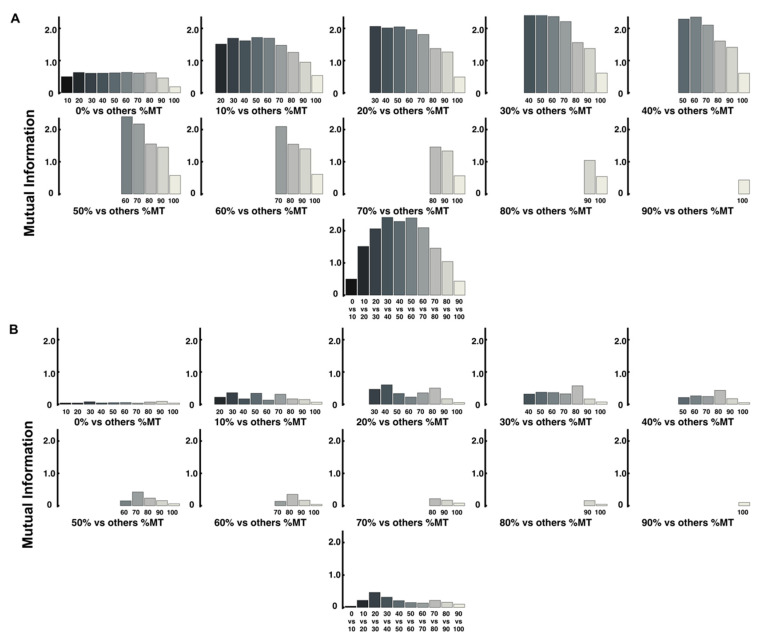
Mutual information between different time points of the trajectory is calculated using the encoded values. (**A**) Mutual information of ID6. The bottom part shows the result of mutual information at adjacent time points. (**B**) Mutual information of ID3. The bottom part shows the result of mutual information at adjacent time points.

**Figure 9 entropy-24-00788-f009:**
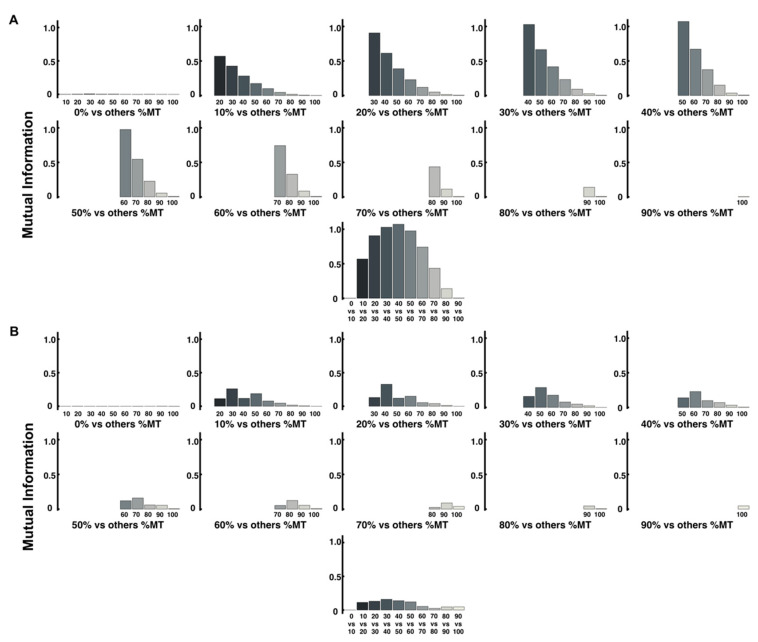
Mutual information between different time points of the trajectory is calculated using the *x*-axis coordinates. (**A**) Mutual information of ID6. The bottom part shows the result of mutual information at adjacent time points. (**B**) Mutual information of ID3. The bottom part shows the result of mutual information at adjacent time points.

**Figure 10 entropy-24-00788-f010:**
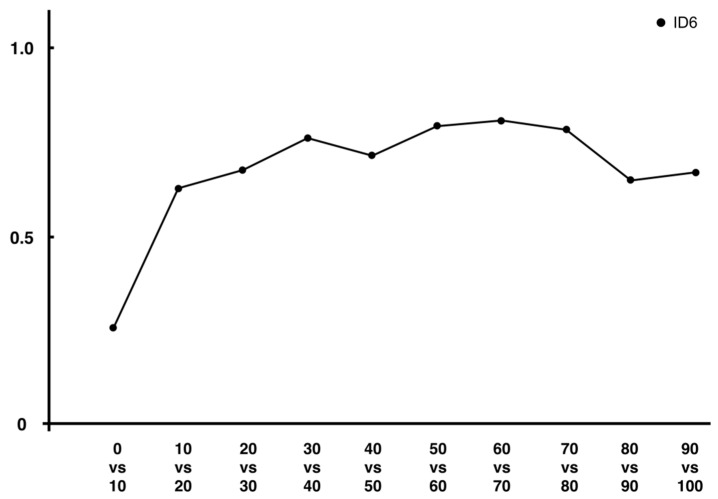
Ratio of mutual information to entropy at a later time point in ID6.

## Data Availability

The datasets generated during and/or analyzed during the current study are available from the corresponding author on reasonable request.

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
