# Peer review of "Estimating Information Processing of Human Fast Continuous Tapping from Trajectories"

_entropy, 2022, doi:10.3390/e24060788_

Round 1

Reviewer 1 Report

  1. There is many more interesting kinds of movement for studing than the tapping by the stylus tip. Why have you chosen this movement?
  2. What is the pixel resolution of the cameras that scan man's movement? The number of camera pixels has an influence on the localization of the moving objects and, thus, the results of computing the trajectories?
  3. Fig. 2 lacks the description of the axes. Almost the same image is in your previous paper Acta Psychologica 220(10), 2021.
  4. Fig. 5B lacks the axis x (not visible).
  5. Line 207: Write "i" rather as a lowercase index. Try to calculate the entropy from the Rényi formula for parameter alpha = 2.
  6. In the text, there are many typographical mistakes. For example, some figure captions are not atteched to the figures.

Reviewer 2 Report

Report on the manuscript entropy-1749084

I have reviewed the manuscript titled “Estimating information processing of human fast continuous tapping from trajectories” aothored by Hiriko Murakami and Norimasa Yamada. The authors proposed to quantify the variability of trajectories during a continuous aiming task under the Fitt empirical paradigm, via quantifiers from Information Theory, such Shannon entropy and mutual information.

The manuscript is well written and structured, and deserves to be published in Entropy. However there are some point that should be clarified.

Minor points:

  • I recommend to increase the numbers in the axes for the sake of better visualization (Fig. 3, 6, 7, 9, and 10).
  • - There is no information about the axes in Fig. 2

Additionally, I would be nice if the authors could get deeper into the following points:

  • Could the entropy be normalized in this context?
  • - The authors said that the entropy is zero if the coordinates of all 100 taps fall within the same cubic space. From Fig. 6 and 7, it is clear that entropy never reaches the zero at the extremes (0 and 100 %), but the width of the target is 15 mm (the same size of the 3D-bin) Why?.
  • Is it possible to define a confident interval representing the zero entropy for each percent moving time?
  • I believe that the procedure to estimate the probability distribution could be explained better.

Round 2

Reviewer 1 Report

Figure 5 has no values on the y,x-axes.

Figures 6-7 has no x-axis.
